# Predictors of health self-management behaviour in Kazakh patients with metabolic syndrome: A cross-sectional study in China

Zhihong Ni[1,2], Yulu Wang[2], Ning Jiang[1], Xiaolian Jiang 🟢[1]*

1 West China School of Nursing, Sichuan University, Chengdu, China, 2 Shihezi University School of Medicine, Shihezi, China

* scujxl16@163.com

## Abstract

### Background

Metabolic syndrome (MS) is common among Muslim patients living in China, most of whom are Kazakh adults. Continuous and effective health self-management plays a critical role in preventing negative health outcomes for individuals with MS. However, Muslim minority patients with MS face many difficulties in actively participating in health self-management, and the factors supporting their successful self-management of MS remain unclear.

### Objective

This study aimed to identify the factors predicting health self-management behaviour among Kazakh MS patients and provide empirical evidence for establishing recommendation guidelines or intervention programmes for health self-management among Muslim minorities.

### Methods

A cross-sectional study was conducted in Xinjiang Province, China, with the use of convenience sampling to explore the current health self-management behaviour of 454 Kazakh MS patients and its influencing factors. Univariate analysis and logistic regression were used to analyse the data.

### Results

The total health self-management behaviour score of Kazakh MS patients was 85.84 ±11.75, and the weaknesses in self-management behaviour were mainly reflected in three dimensions: disease self-monitoring, emotion management and communication with physicians. The significant positive predictors of health self-management behaviour were sex, education, family monthly income per capita, weight, knowledge of MS, and self-efficacy, while the significant negative predictors were blood pressure, the number of MS components, chronic disease comorbidities, and social support (objective support and utilization of support).

**Data Availability Statement:** All relevant data are within the paper and its Supporting Information files.

**Funding:** The authors received no specific funding for this work.

**Competing interests:** The authors have declared that no competing interests exist.

## Conclusion

The health self-management behaviour of Kazakh MS patients is poor. Health care providers should aim to develop culturally specific and feasible health management intervention programmes based on the weaknesses and major modifiable influencing factors in Muslim minority MS patient health self-management, thus improving the health outcomes and quality of life of patients.

## 1. Introduction

Metabolic syndrome (MS) is a pathological state in which multiple metabolic risk factors are present in the same individual and manifests mainly as central obesity, dyslipidaemia, hypertension, decreased glucose tolerance or type 2 diabetes [1]. As a high-risk factor and pre-disease status for various chronic diseases such as cardiovascular disease (CVD) and diabetes, MS has become a major public health problem worldwide. The considerable changes that have occurred in people's lifestyles, such as unhealthy diets, sedentary behaviour, and greatly increased psychological pressure, are principally responsible for the increasing onset and spread of MS each year [2–8]. The prevalence of MS in the United States increased from 25.0% in 2006 to 34.2% in 2012 [2]; the prevalence of MS in China increased from 21.3% in 2009 to 33.9% in 2010, and there were approximately 454 million patients with MS nationwide [9]. The prevalence of MS in China had noticeable regional differences, with the prevalence being higher in the north than in the south (23.3% vs. 11.5%) and with adults in Xinjiang (especially Kazakh adults) showing a high prevalence relative to the national average. By 2012, the prevalence of MS in the Kazakh population was as high as 27.7% (36.1% in males and 16.7% in females) and showed an increasing trend with age [10–13].

The various components of MS are interrelated and aggregated, forming the foundation for cardiovascular diseases, diabetes, renal damage, and other chronic diseases; in addition, they are mutually causal, forming a vicious circle that seriously damages people's health, reduces quality of life, and causes a heavy economic burden. The occurrence of MS is related to many factors, such as race, genetics, culture, religious beliefs, and lifestyle [14]. It is generally believed that ethnic/racial minorities have fewer healthcare experiences, poorer health status, and significantly shorter life expectancy than people living in developed regions [15]. Kazakhs are mainly followers of Islam. In China, Kazakhs are nomadic people who have lived mainly in the cold and dry northern alpine regions and the remote, less developed areas of Xinjiang for generations. Long-term nomadic life has caused Kazakhs to develop a unique lifestyle and cultural background that is distinct from other ethnic groups, which may be the main reason for their high incidence of MS [10, 13].

Primary care is often a first step in the treatment of chronic disease, and continuous and effective health self-management is the first line of defence for MS prevention. Health self-management based on the health belief model, self-efficacy theory and knowledge-attitude-belief-practice theory can help patients understand their health status; identify existing and potential health problems; effectively adjust their cognition, behaviour and psychological status; prevent the occurrence and development of diseases; and ultimately, promote their health [16–19].

China is a multi-ethnic country. Due to the substantial differences in economy, culture, and lifestyle across the country, it is a challenge for ethnic minority patients with MS to actively participate in health self-management. It is still unknown whether the current situation and

influencing factors of the health self-management of Kazakh patients with MS in China are similar to those of other countries and ethnic groups. Therefore, this study aimed to explore the current situation of health self-management of Kazakh MS patients in traditional primary health care settings, identify and analyse factors predicting health self-management behaviour, and provide scientific empirical evidence for establishing recommendation guidelines or intervention programmes for the health self-management of Muslim minorities.

## 2. Methods

### 2.1. Study design and subjects

A cross-sectional design was employed in this study. From December 2017 to June 2018, 454 Kazakh patients with MS from three primary health centres in Qingshuihe Township, Chaichang Village and Huosiaerke Village, Xinjiang, China, were recruited through the convenience sampling method. The three study sites are all typical Kazakh ethnic communities and representative of the Kazak people in Xinjiang in terms of the historical evolution of pastoral areas, the production activities of herdsmen, lifestyles, customs, and economic levels. The inclusion criteria for participants were as follows: the Kazakh population had residential status (a resident for more than 6 months), was aged 18–70 years old, met the International Diabetes Federation (IDF) diagnostic criteria for MS, provided informed consent and participated voluntarily. The exclusion criteria were as follows: patients who were migrants, were pregnant women, had serious chronic disease comorbidities, had a history of cognitive impairment or psychiatric illness, had impaired hearing and/or vision, were unable to communicate properly, refused to participate in this study or were currently participating in other studies.

### 2.2. Sample size

This study adopts regression analysis as the main statistical analysis method, and health self-management behaviour as the dependent variable. It is generally believed that the sample size should be 10~20 times the number of independent variables. There were 27 independent variables in this study. Thus, the sample size was 15 times the number of independent variables, 405 cases, which was increased by 20% to a total of 486 cases considering the possibility of non-response, data loss and sample loss.

### 2.3. Measurement indicators and tools

In this study, the content validity of each scale was determined by an expert panel of researchers involved in chronic metabolic disease, cardiovascular disease, and chronic disease management; Cronbach's α and test-retest reliability were determined by the pre-survey results (the surveys were conducted on the 1st and 14th days of the pre-survey period).

**2.3.1. General information questionnaire.** Based on the study objectives and the analysis of relevant literature, we designed the general information questionnaire, which included (1) general demographic data: sex, age, education, marital status, occupation, residential status, family monthly income per capita, and payment method of medical expenses; (2) disease-related data: history of chronic disease and family genetic history; and (3) basic disease data: weight, waist circumference (WC), body mass index (BMI), blood pressure (BP), fasting plasma glucose (FPG) and blood lipids, all measured using internationally standardised methods. Overweight and obesity were defined as a BMI≥24.0 kg/m2 and a BMI≥28.0 kg/m2, respectively.

**2.3.2. Knowledge of MS (K-MS) Scale.** The K-MS scale was developed by See et al. [20] and comprises 10 items in 3 subscales (all single choice with 5 options per question): the

definition of MS, the relationship between MS and CVD, and the prevention of MS. The scoring method involves calculating the number of correct answers, with each correct answer counting for 10 points; the total score ranges from 0–100, with higher scores indicating better knowledge of MS. The content validity index of the scale (S-CVI) was 0.98. In this study, Cronbach's α was 0.90, and the test-retest reliability was 0.88.

**2.3.3. Chronic disease self-efficacy scale.**   This scale was developed by Lorig et al. [21] and comprises six items in two dimensions: general disease management and symptom management. The scale uses a 10-level scoring method, with each item scoring 1–10 points and the total score being the mean of the six items. Higher scores indicate higher self-efficacy. The S-CVI of the Chinese version of the scale was 0.92. In this study, Cronbach's α was 0.93, and the test-retest reliability was 0.87.

**2.3.4. Self-Management Behaviour of MS patients (SMB-MS) scale.**   The SMB-MS Scale was modified and supplemented by the researchers based on the self-management theory of chronic diseases [22], with the national chronic disease self-management program study questionnaire [21] as a template, as well as related MS patient health self-management assessment tools [23, 24], the Summary of Diabetes Self-Care Activities (SDSCA) [25] and the analysis of literature on health self-management of Kazakh MS patients in Xinjiang. The modified scale includes 36 items in seven dimensions: diet management, exercise management, other lifestyle management (including the management of sleeping, sedentary, smoking/passive smoking and alcohol consumption), medication management, disease self-monitoring, emotion management, and communication with physicians. The scale uses a 5-item Likert scoring scale, with each item scoring 1–5 points; the total score ranges from 36–180, with higher scores indicating better self-management behaviour of MS patients. In this study, the S-CVI was 0.98, the item CVI (I-CVI) was 0.80~1.00, Cronbach's α was 0.87, and the test-retest reliability was 0.96.

**2.3.5. Social Support Rating Scale (SSRS).**   The SSRS was developed by Xiao [26] and comprises 10 items in three dimensions: subjective support, objective support, and the utilization of support. The scale uses a positive cumulative scoring method, with the total score derived from the scores of the 10 items and a maximum score of 66, with higher scores indicating better support. The I-CVI of the scale was 0.89~0.94. In this study, Cronbach's α was 0.72, and the test-retest reliability was 0.76.

## 2.4. Data collection

Unified training was conducted with the data collectors to ensure consistency in Kazakh language translation and understanding of the questionnaire with the Chinese questionnaire. In principle, the questionnaires should have been completed by the respondents themselves. However, for respondents with reading or writing difficulties, the investigator assisted them item-by-item using neutral, non-suggestive language (Chinese/Kazakh). To ensure the integrity, authenticity, and accuracy of the data, during the on-site completion of the questionnaire, the respondents were asked to check and complete any items that were missing or in doubt, and then the questionnaires were collected after verification. Anthropometric and physiological data collection were performed strictly in accordance with the specimen collection specifications, and the relevant instruments were calibrated before each use.

## 2.5. Data analysis

EpiData 3.1 software was used for double data entry, logical checks and random extraction of 5% of the data review for strict control of the data entry quality. The data were statistically analysed using SPSS 22.0 software. The count data were described by the frequencies and composition ratios. The normal distribution of the measurement data was described as the mean

±standard deviation, while the skewed distribution was described as the median and inter-quartile range. Logistic regression analysis explored the influencing factors of health self-management behaviour in Kazakh MS patients in China.

## 2.6. Ethical permission

This study strictly followed the biomedical ethics code and was approved by the West China Hospital of Sichuan University biomedical research ethics committee (Approval No. 2017 (389)). The study was conducted after the subjects agreed and signed the informed consent form.

## 3. Results

### 3.1. Sample characteristics

**3.1.1. Demographic characteristics.** The participants were all Muslim Kazakhs with an average age of 49.92±12.07 years old; 52.2% of the participants were women. Residents living in pastoral areas accounted for 91.4% of the participants. A total of 85.7% of the participants were married, and 10.6% lived alone. The participants mainly had a primary school education (37.9%) or were illiterate (25.1%); agricultural and livestock workers accounted for 96.7% of the participants. The participants' medical expenses were mainly covered by urban medical insurance, accounting for 89.9%, and the family income per capita of 70.0% of the participants was 1000 yuan or less (Table 1).

**3.1.2. Disease characteristics.** The mean BMI of the participants was 28.48±4.03 kg/m$^2$ (28.68±3.49 kg/m$^2$ for males and 28.30±4.46 kg/m$^2$ for females); their mean weight was 77.48 ±14.05 kg (84.61±12.25 kg for males and 70.95±12.33 kg for females), with 37.9% being over-weight and 49.1% being obese. According to the IDF diagnostic criteria for MS, WC and BMI showed the most prominent abnormal rates among all the components of MS (100% of males with a systolic BP (SBP)≥130 mmHg and 94.1% of females with a WC≥80 cm), and the major-ity of patients (78.4%) had three MS components. A total of 312 participants (68.7%) had chronic disease comorbidities, of which rheumatoid arthritis represented the majority (38.8% of the total), followed by hypertension (38.1% of the total); 51.5% of the participants reported no family history of MS (Fig 1 and Table 1).

### 3.2. Health self-management behaviour

The total health self-management behaviour score of the participants was 85.84±11.75 points and ranged from 53 to 121 points. Among the seven dimensions, the exercise management dimension had the highest item mean score (2.82±0.66), and the disease self-monitoring dimension had the lowest item mean score (1.48±0.43). Weaknesses in the self-management behaviour of the participants were mainly reflected in three dimensions: disease self-monitor-ing, emotion management, and communication with physicians (Table 2).

### 3.3. Factors predicting health self-management behaviours in MS patients

**3.3.1. Univariate analysis of health self-management behaviour.** Adopting the nonpara-metric test (since most of the independent variables could not achieve homogeneity of variance of the dependent variable at different levels of grouping, P <0.05), the analysis results showed the following (Table 1): there were statistically significant differences in the health self-man-agement behaviour of Kazakh MS patients in terms of age, family monthly income per capita, and chronic disease comorbidities (P<0.05).

**Table 1. Univariate analysis of general characteristics and self-management behaviour of Kazakh MS patients (n = 454).**

| Variables | Patients (%) | Self-management behaviour | $\chi^2$/ Z | P |
|---|---|---|---|---|
| **Age (year)** | | | | |
| ≤40 | 111 (24.5) | 81.61±9.59 | 26.782¶ | <0.001* |
| 40~60 | 253(55.7) | 88.34±11.72 | | |
| ≥60 | 90(19.8) | 84.02±12.50 | | |
| **Sex** | | | | |
| Male | 217(47.8) | 87.31±12.92 | -1.609# | 0.108 |
| Female | 237(52.2) | 84.89±10.40 | | |
| **Education** | | | | |
| Illiteracy | 114(25.1) | 83.10±9.24 | 1.425¶ | 0.700 |
| Elementary school | 172(37.9) | 85.59±12.50 | | |
| Middle school | 128(28.2) | 87.51±11.36 | | |
| High school and above | 40(8.8) | 86.58±11.98 | | |
| **Occupation** | | | | |
| Agriculture and animal husbandry | 439(96.7) | 85.92±11.77 | -1.032# | 0.302 |
| Non-agriculture and animal husbandry | 15(3.3) | 83.47±11.06 | | |
| **Marital Status** | | | | |
| Partnered | 389(85.7) | 85.75±11.77 | -0.962# | 0.336 |
| Un-partnered | 65(14.3) | 86.33±11.69 | | |
| **Living status** | | | | |
| Live alone | 48(10.6) | 85.44±11.61 | -0.385# | 0.700 |
| Live with others | 406(89.4) | 85.88±11.76 | | |
| **Place of residence** | | | | |
| Pastoral area | 415(91.4) | 85.97±11.71 | -0.964# | 0.335 |
| Cities and towns | 39(8.6) | 84.44±12.19 | | |
| **Method of paying medical expenses** | | | | |
| Private expense | 46(10.1) | 83.04±10.93 | -1.280# | 0.200 |
| Urban medical insurance | 408(89.9) | 86.15±11.81 | | |
| **Income (yuan/month/person)** | | | | |
| ≤1000 | 318(70.0) | 85.85±11.81 | 9.007¶ | 0.029* |
| 1001~3000 | 83(18.3) | 84.24±11.21 | | |
| 3001~5000 | 43(9.5) | 86.98±12.28 | | |
| ≥5001 | 10(2.2) | 93.90±8.81 | | |
| **Chronic disease comorbidities** | | | | |
| No | 142(31.3) | 84.46±10.42 | 2.791# | 0.043* |
| Yes | 312(68.7) | 86.47±12.27 | | |
| **Family heredity history** | | | | |
| No | 234(51.5) | 86.11±12.09 | -0.834# | 0.404 |
| Yes | 220(48.5) | 85.55±11.39 | | |
| **Number of MS components** | | | | |
| 3 | 356(78.4) | 86.28±11.93 | 2.457¶ | 0.293 |
| 4 | 81(17.8) | 83.98±11.23 | | |
| 5 | 17(3.7) | 85.41±9.72 | | |

Note: #, Mann-Whitney U 检验

¶, Kruskal-Wallis H 检验

*, P<0.05.

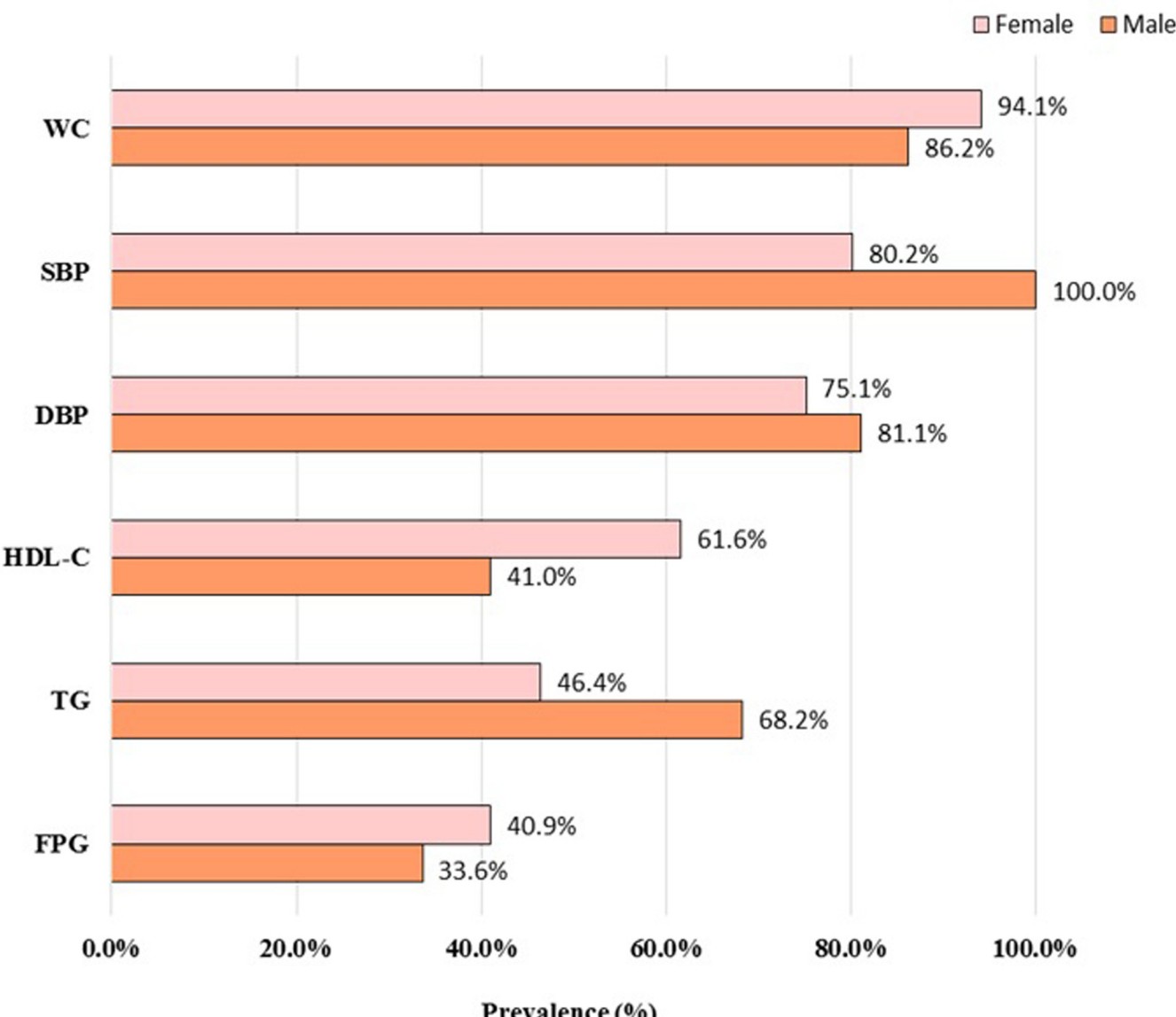

**Fig 1. Total number of Kazakh patients with each MS components (IDF diagnostic criteria).** Note: WC, waist circumference; SBP, systolic blood pressure; DBP, diastolic blood pressure; HDL-C, high density lipoprotein cholesterol; TG, triglyceride; FPG, fasting plasma glucose.

**3.3.2. Correlation analysis of health self-management behaviour with metabolic indicators, knowledge of MS, self-efficacy and social support.** Adopting Spearman correlation analysis, the results showed the following (S1 and S2 Tables): positive correlations between weight and communication with physicians ($r = 0.111$, $P<0.05$), high density lipoprotein cholesterol (HDL-C) and disease self-monitoring ($r = 0.098$, $P<0.05$), knowledge of MS and other lifestyle management ($r = 0.154$, $P<0.01$) as well as medication management ($r = 0.102$, $P<0.05$), self-efficacy and medication management ($r = 0.096$, $P<0.05$), subjective support and emotion management ($r = 0.103$, $P<0.05$); negative correlations between SBP and diastolic blood pressure (DBP) and medication management ($r = -0.104$, $P<0.05$ and $r = -0.130$, $P<0.01$), knowledge of MS and exercise management ($r = -0.172$, $P<0.001$) as well as disease

**Table 2. Overall health self-management behaviour scores of Kazakh MS patients (*n* = 454).**

| Variables | Mean ± SD (scale) | Range | Scale range | Mean ± SD (item) | Item range |
|---|---|---|---|---|---|
| **Overall SMB** | 85.84±11.75 | 53–121 | 36–180 | 2.38±0.33 | 1–5 |
| Exercise management | 8.47±1.98 | 3–12 | 3–15 | 2.82±0.66 | 1–5 |
| Other lifestyles management | 11.23±3.31 | 4–19 | 4–20 | 2.81±0.83 | 1–5 |
| Diet management | 33.59±5.03 | 20–44 | 12–60 | 2.80±0.42 | 1–5 |
| Medication management | 7.76±2.79 | 3–14 | 3–15 | 2.59±0.93 | 1–5 |
| Communication with physicians | 6.02±2.77 | 3–13 | 3–15 | 2.01±0.92 | 1–5 |
| Emotion management | 11.37±3.47 | 6–23 | 6–30 | 1.90±0.58 | 1–5 |
| Disease self-monitoring | 7.39±2.16 | 5–16 | 5–25 | 1.48±0.43 | 1–5 |

Note: SMB, self-management behaviour; SD, standard deviation.

self-monitoring (*r* = -0.102, *P*<0.05), and utilization of support and other lifestyle management (*r* = -0.097, *P*<0.05).

**3.3.3. Multi-factor analysis of health self-management behaviour.** The total health self-management behaviour score and each dimension score were taken as the dependent variables, and the statistically significant variables in the above univariate analysis results as well as the nonsignificant variables that were still judged to be meaningful from a professional perspective were taken as the independent variables. Logistic regression analysis was conducted (forward logistic regression, $a_{entry}$ = 0.05, $a_{out}$ = 0.10), and the specific variables and assignments are shown in S3 Table.

The results of regression analysis showed the following (Table 3): sex (male), education, family monthly income per capita, weight, knowledge of MS, and self-efficacy were positive influencing factors of health self-management behaviour, whereas disease characteristics (DBP, number of MS components, and chronic disease comorbidities) and social support (objective support and utilization of support) were negative influencing factors of health self-management behaviour.

## 4. Discussion

To our knowledge, the present study is a more comprehensive recent cross-sectional study examining the current state of health self-management behaviours and their influencing factors in patients with metabolic syndrome. These data provide essential evidence for more accurate and targeted intervention studies by those involved in the health management of this population.

Several studies have shown that the current status of health self-management behaviours in MS patients is generally less than ideal, and there are very few reports on the influencing factors of health self-management behaviours in MS patients. Health self-management behaviours are central to evaluating patient health outcomes, and it is essential to understand their current status and influencing factors. Based on this, we conducted a further study.

### 4.1. Current status of health self-management behaviour in Kazakh MS patients

The total SMB-MS Scale scores ranged from 36–180 points. In this study, the mean health self-management behaviour score of the participants was 85.84±11.75 points and ranged from 53 to 121 points, with only 32 participants (7.5%) scoring the highest possible score of 60% or higher (108 points); these results indicate that the overall health self-management behaviour of

**Table 3. Multi-factor analysis of health self-management behaviour (*n* = 454).**

| Variables | Predictor | B | SE | P | OR | OR 95% CI |
|---|---|---|---|---|---|---|
| Overall SMB | TC | 0.460 | 0.181 | 0.011 | 1.585 | 1.112, 2.258 |
| | Objective support | 0.192 | 0.086 | 0.026 | 1.211 | 1.023, 1.434 |
| | Utilization of support | 0.164 | 0.084 | 0.042 | 1.179 | 0.999, 1.391 |
| | Weight | -0.035 | 0.013 | 0.007 | 0.966 | 0.942, 0.991 |
| | Symptom management self-efficacy | -0.170 | 0.094 | 0.041 | 0.844 | 0.701, 1.015 |
| Diet management | Occupation | 1.264 | 0.548 | 0.021 | 3.540 | 1.209, 10.36 |
| | Number of MS components | 0.474 | 0.237 | 0.045 | 1.606 | 1.010, 2.555 |
| | General disease management self-efficacy | -0.149 | 0.051 | 0.003 | 0.862 | 0.780, 0.952 |
| | TG | -0.159 | 0.093 | 0.006 | 0.853 | 0.711, 1.023 |
| Exercise management | Marital Status | 0.565 | 0.275 | 0.040 | 1.759 | 1.026, 3.014 |
| | Family heredity history | -0.488 | 0.213 | 0.022 | 0.614 | 0.404, 0.932 |
| | Income (yuan/month/person) | -0.670 | 0.330 | 0.043 | 0.512 | 0.268, 0.978 |
| Other lifestyles management | Income (yuan/month/person) | 0.826 | 0.348 | 0.018 | 2.284 | 1.154, 4.520 |
| | Objective support | 0.070 | 0.032 | 0.029 | 1.072 | 1.007, 1.142 |
| | Knowledge of MS | -0.020 | 0.009 | 0.022 | 0.980 | 0.964, 0.997 |
| | Chronic disease comorbidities | -0.808 | 0.233 | 0.001 | 0.446 | 0.282, 0.704 |
| | Education | -2.140 | 0.671 | 0.001 | 0.118 | 0.032, 0.438 |
| Medication management | DBP | 0.035 | 0.015 | 0.020 | 1.036 | 1.005, 1.067 |
| | Utilization of support | -0.091 | 0.041 | 0.026 | 0.913 | 0.843, 0.989 |
| | Income (yuan/month/person) | -0.353 | 0.133 | 0.008 | 0.702 | 0.541, 0.912 |
| | Education | -0.512 | 0.219 | 0.019 | 0.599 | 0.390, 0.920 |
| Disease self-monitoring | Chronic disease comorbidities | 1.745 | 0.394 | <0.001 | 5.724 | 2.644, 12.391 |
| | Living status | 1.136 | 0.398 | 0.004 | 3.115 | 1.427, 6.796 |
| | Utilization of support | 0.114 | 0.044 | 0.010 | 1.121 | 1.027, 1.222 |
| | Knowledge of MS (definition) | 0.064 | 0.016 | <0.001 | 1.066 | 1.034, 1.100 |
| | Objective support | 0.060 | 0.031 | 0.031 | 1.061 | 1.000, 1.127 |
| | Education | -1.256 | 0.259 | <0.001 | 0.511 | 0.114, 0.833 |
| Emotion management | Chronic disease comorbidities | 0.630 | 0.327 | 0.044 | 1.878 | 0.989, 3.566 |
| | Income (yuan/month/person) | -0.359 | 0.184 | 0.041 | 0.699 | 0.487, 1.002 |
| Communication with physicians | Utilization of support | 0.251 | 0.060 | <0.001 | 1.286 | 1.144, 1.445 |
| | Objective support | 0.100 | 0.047 | 0.032 | 1.105 | 1.009, 1.211 |
| | Weight | -0.026 | 0.010 | 0.006 | 0.974 | 0.956, 0.992 |
| | Sex | -0.558 | 0.258 | 0.031 | 0.572 | 0.345, 0.950 |
| | Number of MS components | -1.134 | 0.435 | 0.009 | 0.322 | 0.137, 0.754 |

Note: B, regression coefficient; SE, standard error; SMB, self-management behaviour; TC, total cholesterol; TG, triglyceride; DBP, diastolic blood pressure.

Kazakh MS patients in China was poor. Moreover, consistent with the study report on the self-management behaviour of diabetes patients by Huang et al. [27], Chinese Kazakh MS patients also showed variation in the mean item scores across the dimensions of health self-management behaviour, with the mean scores for each dimension exhibiting the following order from highest to lowest: exercise management, other lifestyle management, diet management, medication management, communication with physicians, emotion management, and disease self-monitoring. These findings indicated that different regions and groups exhibit differences in health self-management behaviour.

Regarding studies of other populations in China, this study was consistent with the findings of some studies of ethnic minority patients with MS or related diseases. For example, Tang

et al. [28] surveyed rural minority patients with chronic diseases in western China; Cai et al. [29] surveyed Na Xi, Li Shu, Dai and Jing Po hypertension patients in rural southwestern China; Geira [30] surveyed elderly Uygur patients with type 2 diabetes; Yan et al. [31]surveyed Tibetan patients with type 2 diabetes; Su et al. [32] surveyed ethnic minority patients with diabetes in Yunnan Province, all of whom showed significantly poorer overall levels of health self-management behaviours among ethnic minority patients with hypertension, diabetes and other chronic diseases. However, Wang et al.'s survey results showed that the health self-management behaviour of Hui patients with type 2 diabetes was generally at a moderate level [33]. Ge et al.'s survey results showed that the health self-management behaviour of patients with impaired glucose regulation in Guangzhou was generally at the upper-middle level [34], and Huang et al.'s survey results showed that the health self-management behaviour of patients with type 2 diabetes in Chengdu was generally at a good level [27]. These results are partly attributable to the creation of healthy cities in China, an effort that has engendered a supportive environment for the prevention and control of chronic disease and health promotion, as well as the rapidly developing comprehensive prevention and control system for chronic diseases, including diabetes, in urban areas.

The results of this study were consistent with the findings of studies of patients with MS or related diseases in other countries. A qualitative study by Lundberg et al. [35] showed that most Thai Buddhist and Muslim women with type 2 diabetes reported that it was very difficult to change their lifestyles and perform health self-management according to the advice of medical staff, and their overall level of health self-management behaviour was generally poor; a large-scale survey of 19,843 black, white, and Hispanic diabetic patients in the 50 states of the United States by Oster et al. showed that all racial/ethnic groups had low levels of health self-management behaviours and that there were racial and ethnic differences [36]. Here, the explanation may relate to the unique lifestyle and religious beliefs of the studied ethnic groups. Religious taboos in the Quran, such as those regarding diet, behaviour and certain types of emotional expression, have a profound influence on Kazakhs who believe in Islam. These taboos not only play an important guiding role with respect to their health concepts, lifestyles and social life, but also have a strong restraining effect, which may affect the health self-management behaviour of Kazakh MS patients to a certain extent.

## 4.2. Factors predicting health self-management behaviour in Kazakh MS patients

**4.2.1. Sociodemographic factors.** The sociodemographic factors affecting the health self-management behaviour of Kazakh MS patients included sex, education, and family monthly income per capita. Among them, sex had an influence on the communication with physicians dimension; education had an influence on the exercise management, other lifestyle management, medication management, and communication with physicians dimensions; and family monthly income per capita had an influence on the exercise management, other lifestyle management, medication management, and emotional management dimensions.

*4.2.1.1. Sex.* The results of this study indicated that being male was a protective factor among Kazakh MS patients for communication with physicians (Table 3). During the formation and development of the Kazakh nation, family has always been regarded as the basic unit of agricultural and livestock production activities. The traditional nomadic culture, economy, and lifestyle and the harsh natural environment have given special status, responsibility and meaning to the roles and status of Kazakh men in the family and contributed to the formation of the traditional concept of female subordination—of breadwinning men and homemaking women; the social activities of Kazakh women have usually been limited to interactions with

relatives and neighbours. Studies have shown that, due to factors such as the limitations of the material, economic, and cultural conditions and disease cognition, socially disadvantaged individuals tend to engage in harmful behaviours [37], which partly explains the poor communication behaviours between female Kazakh MS patients and physicians. In addition, in the traditional family lifestyle of Kazakh, the daily life and diet of the whole family are managed by women, while men generally do not enter the kitchen to prepare food. Kazakhs traditionally believe that men who enter the kitchen as a sign of the hostess's incompetence in performing her duties and will be ridiculed by others. Therefore, researchers should consider the above particular Kazakh custom when developing dietary self-management interventions.

*4.2.1.2. Education.* The univariate analysis showed that education was significantly associated with the scores for the other lifestyle management and disease self-monitoring dimensions of health self-management behaviour among Kazakh MS patients in China (Table 1). In the logistic regression analysis, education was a significant predictor that determined the health self-management behaviour of MS patients. Education was a protective factor for other lifestyle management behaviours, medication management behaviour and disease self-monitoring behaviour (Table 3). These findings were consistent with the results of Tang et al. [28] on chronic diseases among ethnic minorities in rural western China (mainly Zhuang, Hui, Uygur, and Mongolian ethnicities): low education levels lead to too low levels of health knowledge among rural ethnic Chinese minorities, thus preventing them from seeking health care services. In addition, most of the participants in this study could use Chinese and Kazakh for basic oral communication, but more than 50% of the participants had difficulties reading and writing Chinese and Kazakh because of educational limitations (only primary school education or illiteracy), which restricted their health self-management behaviour to a certain extent. Multiple studies [38–41] have shown that well-educated patients have more pathways to receive health-related information, a higher degree of disease cognition and perception, and a lower possibility of adverse health behaviour; as the education level improves, personal understanding of health-related information and health advice from health care providers and awareness of quality of life increase, which may affect patients' lifestyles, behaviours, psychosocial attitudes, and chances to access health care services, resulting in good health self-management behaviour. This finding suggests that health care providers should explore practical and cost-effective health self-management intervention programmes according to the different education levels of patients, especially for ethnic minority MS patients with low education levels and even language communication difficulties.

*4.2.1.3. Family monthly income per capita.* The univariate analysis showed that family monthly income per capita was significantly associated with the health self-management behaviour among Kazakh MS patients in China (Table 1). In the logistic regression analysis, for Kazakh MS patients, family monthly income per capita was a positive predictor of exercise management behaviour, medication management behaviour and emotion management behaviour, and was a negative predictor of other lifestyle management behaviour (Table 3). These findings were consistent with the meta-analysis results of Luo et al. [42], who found that family income levels were significantly positively correlated with health self-management behaviour and that an increase in family income affected the diet management, medication management and other health self-management behaviours of MS patients. In China, compared with Han people, ethnic minorities are relatively economically disadvantaged, with a low overall education level, a lack of medical resources and a lack of convenient transportation [32]. Patients who are chronically under greater socioeconomic stress generally have less energy for physical exercise and are more inclined to consume low-cost, high-fat, high-calorie foods. This result is associated with the unique production and economic activities of Kazakhs. In general, the frequency of production activities of herdsmen is directly proportional to their

family income. The higher the family income is, the more frequent the production activities. At the same time, as health status directly affects production activities, herdsmen with higher income will pay more attention to their health status, so their medication management behaviour will be relatively good. Furthermore, with frequent production activities, the range of the social life of herdsmen also expands, which could have certain negative effects on their other lifestyle management behaviour, such as sleeping, being sedentary, smoking, and drinking. Therefore, researchers should consider the status of different health self-management behaviours in the context of Kazakh production methods and socioeconomic conditions when targeted guiding the health self-management of Kazakh MS patients.

**4.2.2. Disease characteristics.** In our study, the disease characteristics affecting the health self-management behaviour of Kazakh MS patients in China included weight, DBP, TC, TG, number of MS components, and chronic disease comorbidities (Table 3). Among them, weight was a positive predictor of the overall level of health self-management behaviour and communication with physicians; TG was a positive predictor of diet management behaviour; and the number of MS components was a positive predictor of communication with physicians. TC was a negative predictor of the overall level of health self-management behaviour; DBP was a negative predictor of medication management behaviour; the number of MS components was a negative predictor of diet management behaviour; and chronic disease comorbidities were a negative predictor of disease self-monitoring behaviour and emotion management behaviour.

Correlation analysis and regression analysis showed that the higher the BP was and the more MS components Kazakh MS patients had, the lower the level of health self-management behaviour, mainly in the medication management and communication with physicians dimensions (S1 Table and Table 3), which is consistent with previous studies [27, 39]. These findings may be explained because MS patients with severe abnormal metabolic indicators often need to be treated via lifestyle changes, drugs and other means. However, it is difficult to achieve the desired therapeutic goals for metabolic indicators in a short period of time, and the resulting frustration seriously hinders MS patients' enthusiasm for health self-management. On the other hand, persistent abnormalities in metabolic indicators increase or aggravate chronic disease comorbidities, resulting in a decline in quality of life, further affecting patients' ability for health self-management, thus forming a vicious circle of negative influence between disease states and health self-management behaviour. Studies have shown that the most difficult patients to treat are non-compliant patients who are negative about their health, while medical services are most effective in patients with positive compliance, as they are most likely to accept lifestyle changes and improve adherence to medications, which improves their health condition [43]. Therefore, when conducting health self-management intervention for Kazakh MS patients, researchers must consider the disease characteristics of MS patients, fully mobilize patients' self-perceptions and self-efficacy regarding their health conditions, and help patients gradually achieve their treatment goals and improve their quality of life.

**4.2.3. Knowledge of MS.** The results of our study showed that knowledge of MS was a positive predictor of other lifestyle management behaviour and medication management behaviour and a negative predictor of exercise management behaviour and disease self-monitoring behaviour (S2 Table and Table 3). These findings indicated that Kazakh MS patients' lack of knowledge of MS had a negative impact on lifestyle changes but had no effect on medication management. Kazakh MS patients inherently believed that taking prescription drugs to treat diseases was enough, and thus, they perceived no need and had no willingness to change their behaviour or habits. The above results were consistent with the findings of Alefishatt [44] and Lo et al. [45]. The health belief model suggests that individuals' perceived disease susceptibility and seriousness as well as their perceived benefits and barriers of taking action directly influence their decisions to choose to engage in or maintain health-promoting behaviour. The

results of this study further suggest that patients' lack of knowledge affects their perception of disease hazards and complications and thus has a negative impact on their motivation to change unhealthy behaviour. Therefore, in working with Kazakh MS patients for disease management and the prevention of cardiovascular disease, it is critical to first adopt appropriate educational programmes to improve patients' awareness of MS.

**4.2.4. Self-efficacy.** The results of our study showed that the self-efficacy of Kazakh MS patients was a powerful positive predictor of health self-management behaviour, mainly as reflected in the diet management and medication management dimensions (S2 Table and Table 3), which was consistent with the results of Lo et al. [46]. Self-efficacy is a patients' subjective judgement of their capacity and self-confidence regarding health self-management. Self-efficacy is influenced by patients' own or others' successful experience, alternative experiences, language and emotional incentives when adopting health-promoting behaviours. Studies suggest that beliefs, confidence, and spirituality may be prerequisites for health self-management [47]. In real life, patients often know that lifestyle changes may have a positive impact on their health, but few put them into practice; those with low self-efficacy who prefer to manage their disease through medical workers or drugs are especially unlikely to implement lifestyle changes. This trend suggests that the development of health self-management intervention programmes should be based on self-efficacy theory, starting from the emphasis on helping patients overcome perceived obstacles to behavioural change and providing patients with skills training to enhance self-efficacy, thus achieving the goal of improving health outcomes.

**4.2.5. Social support.** The results of our study showed that different dimensions of social support had different effects on the total health self-management behaviour scores and the scores for each dimension (S2 Table and Table 3). Social support is a psychological or emotional experience in which an individual receives support and help from family, friends, neighbours, colleagues, or other individuals or organizations, which buffers or regulates the stress of negative events on health by regulating the patient's thinking, living habits and social factors, thereby promoting health and improving quality of life. Due to the influence of sociodemographic and cultural characteristics, there are differences in social support and its impact on health self-management behaviour among patients of different ethnicities and races with chronic diseases.

Our study found that Kazakh MS patients generally reported strong social support, especially objective support and utilization of support (S2 Table). The reason is that, on the one hand, the Kazakh nation has traditionally continued to carry out collective and mutual aid production activities and form a relatively stable "awule" (the basic unit in the tribe) based on blood relationships. Herdsmen have a very strong family consciousness and family mutual help. On the other hand, due to the religious beliefs of the Kazakh people, religious thought has deeply penetrated into every aspect of the social life of herdsmen, forming the core of Kazakh traditional culture as well as the inherent values and low-demand, easy-to-satisfy mental state that characterize herdsmen's daily lives and communication. In the context of this long-standing, entrenched approach to social relationships, Kazakh MS patients are generally dependent and have difficulty making decisions or taking action; in addition, they often tend to make many negative assumptions about their health [15]. These considerations elucidate why objective support and utilization of support had negative impacts on some dimensions of health self-management behaviour. These findings suggest that health care providers should base health self-management education for Kazakh MS patients on the health belief model to help patients understand their disease susceptibility, potential harm and possible obstacles in health self-management and guide patients to use social support correctly and effectively to maximize their self-efficacy.

This study has limitations in terms of generalisability of the results. This study was a single-centre study of Kazakh MS patients in Xinjiang, China, with a narrow population and insufficient sample coverage. It is suggested that future research should be conducted in multiple centres with patients from different regions and ethnic groups. A convenience sampling method was used in this study, which may have involved a degree of sampling bias and did not represent the overall situation of Kazakh MS patients in China well. Additionally, due to the limitations of the descriptive study design, only univariate analysis and multivariate logistic regression analysis were conducted in the study, without further exploring the cumulative interactions between the various influencing factors and their mechanisms. Further analytical studies, such as cohort studies, are suggested.

## 5. Conclusion

Kazakh MS patients in China had poor health self-management behaviour overall, and the average scores across the dimensions varied. The highest score was for the exercise management dimension, while the lowest score was for the disease self-monitoring dimension. The main influencing factors included sex, education, family monthly income per capita, disease characteristics, knowledge of MS, self-efficacy, and social support. This study suggested that Kazakh MS patients suffer from a large gap in access to health care and preventive services due to environmental exposure, socioeconomic factors, health behaviours and psychosocial factors. The lack of health-related knowledge and information and the unique traditional cultural context and values constitute the largest obstacles for this particular group when seeking a healthy lifestyle. As a result, in conducting Kazakh MS patient health self-management interventions, researchers should consider the effects of cultural differences and disease characteristics on health self-management behaviour based on the health belief model, starting by increasing patients' disease perception and cognition to stimulate the patient's self-efficacy and then conducting targeted interventions for the weaknesses in health self-management behaviour, thus achieving the goal of improving health outcomes and quality of life.

## Supporting information

**S1 Table. Correlation analysis of health self-management behaviour with metabolic indicators ($n$ = 454).**
(DOCX)

**S2 Table. Correlation analysis of health self-management behaviour with knowledge of MS, self-efficacy and social support ($n$ = 454).**
(DOCX)

**S3 Table. Variables and the assignments of logistic regression analysis.**
(DOCX)

## Acknowledgments

We thank all MS patients and their families for their participation and every member of the research team for their solidarity, trust and cooperation.

## Author Contributions

**Data curation:** Zhihong Ni, Yulu Wang, Ning Jiang.

**Formal analysis:** Zhihong Ni.

**Investigation:** Zhihong Ni, Yulu Wang, Ning Jiang.

**Methodology:** Zhihong Ni.

**Project administration:** Zhihong Ni.

**Resources:** Zhihong Ni.

**Supervision:** Zhihong Ni, Xiaolian Jiang.

**Validation:** Zhihong Ni, Xiaolian Jiang.

**Writing – original draft:** Zhihong Ni.

**Writing – review & editing:** Zhihong Ni.

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
