## [Decision Letter · Decision Letter 0]

29 Sep 2021

PONE-D-20-30645Predictors of health self-management behaviour in Kazakh patients with metabolic syndrome: a cross-sectional study in ChinaPLOS ONE

Dear Dr. Jiang,

Thank you for submitting your manuscript to PLOS ONE. After careful consideration, we feel that it has merit but does not fully meet PLOS ONE’s publication criteria as it currently stands. Therefore, we invite you to submit a revised version of the manuscript that addresses the points raised during the review process.

We look forward to receiving your revised manuscript.

Kind regards,

Rubeena Zakar, Ph.D

Academic Editor

PLOS ONE

Journal Requirements:

4. In the methods section please specify the anthropometric and physiological variables studies, and how these variables were measured.

5. Please complete all items on the Clinical Studies Checklist that are relevant for your submission, by following this link: http://journals.plos.org/plosone/s/file?id=dc11/PLOSOne_Clinical_Studies_Checklist.docx (Contact us at plosone@plos.org if you cannot access the document.) There may be overlap between the checklist items and other queries listed below; please address any duplicated queries both in your response email and on the checklist itself. Upload the completed Clinical Studies Checklist as file type “Other” when you re-submit your manuscript. This document is for internal journal use only and will not be published if your article is accepted. The requested information will help us to assess whether your submission complies with PLOS ONE’s policies and adheres to applicable reporting standards. Note that your manuscript may be rejected if you provide incomplete or inadequate responses to the checklist questions and that changing the ‘Section/Category’ of your article does not affect this requirement.

Reviewers' comments:

Reviewer's Responses to Questions

**Comments to the Author**

1. Is the manuscript technically sound, and do the data support the conclusions?

Reviewer #1: Yes

Reviewer #2: Yes

2. Has the statistical analysis been performed appropriately and rigorously? 

Reviewer #1: Yes

Reviewer #2: Yes

3. Have the authors made all data underlying the findings in their manuscript fully available?

Reviewer #1: Yes

Reviewer #2: Yes

4. Is the manuscript presented in an intelligible fashion and written in standard English?

Reviewer #1: Yes

Reviewer #2: Yes

5. Review Comments to the Author

Reviewer #1: I have attached separately anyway these comments.

Review:

‘Predictors of health self-management behaviour in Kazakh patients with metabolic syndrome: a cross-sectional study in China’

In my opinion, this is a well-developed paper. The authors present clearly the problem addressed and go thoroughly through it. I also consider that it represents a contribution in terms of incorporating culturally specific issues that should be taken into account when designing health interventions.

I have the following comments, which I consider important to improve in this paper.

1. The ‘Other lifestyle management” dimension is a broad category. Since it is significant in different areas, e.g., education, I would suggest giving a couple of examples to illustrate what is included under ‘other lifestyle management’.

2. ‘Communication with physicians’: I would suggest to specify whether Kazakh people speak their own language, or whether they are linguistically integrated. Since more than 50% of the participants registered a low educational level (primary school education or illiterate), this could be exacerbated by language issues. If so, this issue should also be taken into account in conducting patient health self-management interventions.

3. [Lines 235-240] Would it be possible to include some information which may explain these results (‘moderate level’, ‘upper-middle level’, ‘good level’) in other ethnic groups? In order to understand what is it about those studies that show different results.

4. The authors state at the beginning that Kazakh people are Muslims, but this dimension is not addressed in the paper.

Besides the arguments given on how traditional Kazakh culture affects health self-management behaviour, it would be useful to know whether there are factors which may be directly related with their religion, e.g., diet. It might be easier to change other areas which may affect HSM, but religious belief that guide daily life are more difficult to address, and therefore it is relevant to identify them; for example, the Lundberg study (cited in lines 242-245): ‘Muslim women with type 2 diabetes reported that it was very difficult to change their lifestyles and perform health self-management according to the advice of medical staff’’.

5. Though the authors declare that their methodological approach will be based on univariate analysis, in order to fully understand the impact of the three dimensions, it would be necessary to present data on the interactions among gender (1), education (2), and family monthly income per capita (3). Or at least between gender and education.

Secondly, how these three variables interact with ‘Knowledge of MS’ (4.2.3), ‘Self-efficacy (4.2.4)’, and ‘Social support’ (4.2.5)

Otherwise, we don`t know whether women have poor communication with doctors just because they are women or because they also have lower educational levels than men.

6. The analysis undergone in the Social support section (4.2.5) is quite well accomplished since it examines the cultural causes and how they relate, as well as explaining its relationship with health self-management behaviour .

7. [Lines 420-421] ‘…researchers should consider the effects of cultural differences and disease characteristics on health self-management behaviour based on the health belief model’.

It is quite valuable that the authors conclude highlighting the effects of cultural differences, but precisely to do so, authors should be more specific on the points suggested above.

8. Though I recognise that it is beyond the authors’ methodological scope, I would suggest presenting at least one DAG, which will allow greater understanding when designing interventions, since it would illustrate interactions among the different studied variables.

9. [Line 159] ‘mean BMI of the participants was 8.48’. Is it 28.4 instead of 8.4?

10. Reference 23:

23. Jaqueline, Vicente E, María Isabel, et al. Cuestionario de asertividad centrado en el estilo de vida CACEV) en pacientes con sindrome metabolico: desarrolloy validacion. Behavioral Psychology /Psicología Conductual. 2017;25(2):349-69.

(a) There is a confusion with the Spanish names and surnames:

The correct way is: Garcia-Silva J, Caballo VE, Peralta-Ramírez MI, Lucena-Santos P, Navarrete-Navarrete N.

 Garcia-Silva J, Caballo VE, Peralta-Ramírez MI, et al.

(b) ‘desarrolloy validacion’ should say: desarrollo y validacion

Reviewer #2: The article presents a very important and relevant topic for public health. I really liked the methodology and discussion. My only suggestion is to improve the clinical and practical implications of the study.

6. PLOS authors have the option to publish the peer review history of their article (what does this mean?). If published, this will include your full peer review and any attached files.

Reviewer #1: **Yes: **Dr. Josiane Bonnefoy

Reviewer #2: **Yes: **Mateus Dias Antunes

---

## [Author Response · Author response to Decision Letter 0]

4 Jan 2022

Response to Reviewers

Dear Editors and Reviewers:

We would like to thank you for your kind letter and for the reviewers’ constructive comments concerning our manuscript entitled “Predictors of health self-management behaviour in Kazakh patients with metabolic syndrome: a cross-sectional study in China” (Manuscript No.:PONE-D-20-30645). All the comments were valuable and highly helpful in revising and improving our paper as well as ensuring the significance of our research was clear. All the authors seriously discussed all the comments. According to the reviewers’ comments, we have made corrections that we hope will be met with approval. The revised sections are marked in red in the paper. The main corrections in the paper and our responses to the reviewer’s comments are as follows:

Responds to the reviewer’s comments:

Reviewer #1: 

1. Comment: “suggest giving a couple of examples to illustrate what is included under ‘other lifestyle management’”.

Response: 

We apologize for neglecting the description of the category of "other lifestyle management" in the manuscript. We have now provided a supplementary explanation in lines 110-111.

2. Comment: “suggest to specify whether Kazakh people speak their own language, or whether they are linguistically integrated”.

Response: 

Considering the reviewer’s suggestion, we have provided specific explanations in lines 300-304.

3. Comment: “[Lines 235-240]: explain these results (‘moderate level’, ‘upper-middle level’, ‘good level’) in other ethnic groups”.

Response: 

Considering the reviewer’s suggestion, we have provided the explanations in lines 241-245. 

4. Comment: “it would be useful to know whether there are factors which may be directly related with their religion, e.g., diet”.

Response: 

As the reviewer suggested, the health self-management behaviour of Kazakh MS patients is directly related to their religious beliefs, which are explained in lines 409-412. Additionally, we explain the correlation with diet and other aspects in lines 253-258.

5. Comment: “suggest to present data on the interactions among gender, education, and family monthly income per capita”.

Response: 

Thank you very much for your suggestion. It is in fact necessary to further explore the interaction of various influencing factors on health self-management behaviour. However, due to the limited length of the article, only univariate analysis and multivariate logistic regression analysis were conducted. We intend to further analyse this topic elsewhere (i.e., in research on the influencing mechanism of health self-management behaviour in Kazakh patients with metabolic syndrome).

In addition, the reasons for “women have poor communication with doctors” are explained in lines 275-282.

6. Comment: “The analysis undergone in the Social support section (4.2.5) is quite well accomplished since it examines the cultural causes and how they relate, as well as explaining its relationship with health self-management behaviour”.

Response: 

Thank you very much for this appreciative comment.

7. Comment: “consider making more specific suggestions on cultural differences”.

Response: 

This problem is described in lines 430-434.

8. Comment: “suggest presenting at least one DAG”.

Response: 

Thank you very much for your advice. In response to your suggestion, we will consider presenting DAG in another study (i.e., in research on the influencing mechanism of health self-management behaviour in Kazakh patients with metabolic syndrome) to better clarify the interactions among the different studied variables.

9. Comment: “[Line 159] ‘mean BMI of the participants was 8.48’. Is it 28.4 instead of 8.4?”.

Response: 

We sincerely regret our careless error. Thank you very much for noting this matter. We have corrected the mistake according to your suggestions.

10. Comment: “Reference 23, there is a confusion with the Spanish names and surnames”.

Response: 

Thank you very much for your comment. We have amended the text accordingly.

Reviewer #2: 

Comment: “The article presents a very important and relevant topic for public health. I really liked the methodology and discussion. My only suggestion is to improve the clinical and practical implications of the study”.

Response: 

Thank you very much for your positive comments on our manuscript. We have tried our best to improve the manuscript according to your suggestion.

---

## [Decision Letter · Decision Letter 1]

18 Apr 2022

PONE-D-20-30645R1Predictors of health self-management behaviour in Kazakh patients with metabolic syndrome: A cross-sectional study in ChinaPLOS ONE

Dear Dr. Jiang,

Thank you for submitting your manuscript to PLOS ONE. After careful consideration, we feel that it has merit but does not fully meet PLOS ONE’s publication criteria as it currently stands. Therefore, we invite you to submit a revised version of the manuscript that addresses the points raised during the review process.

We look forward to receiving your revised manuscript.

Kind regards,

Rubeena Zakar, Ph.D

Academic Editor

PLOS ONE

Journal Requirements:

Reviewers' comments:

Reviewer's Responses to Questions

**Comments to the Author**

1. If the authors have adequately addressed your comments raised in a previous round of review and you feel that this manuscript is now acceptable for publication, you may indicate that here to bypass the “Comments to the Author” section, enter your conflict of interest statement in the “Confidential to Editor” section, and submit your "Accept" recommendation.

Reviewer #2: All comments have been addressed

Reviewer #3: All comments have been addressed

Reviewer #4: (No Response)

2. Is the manuscript technically sound, and do the data support the conclusions?

Reviewer #2: Yes

Reviewer #3: Yes

Reviewer #4: Partly

3. Has the statistical analysis been performed appropriately and rigorously? 

Reviewer #2: Yes

Reviewer #3: Yes

Reviewer #4: No

4. Have the authors made all data underlying the findings in their manuscript fully available?

Reviewer #2: Yes

Reviewer #3: Yes

Reviewer #4: Yes

5. Is the manuscript presented in an intelligible fashion and written in standard English?

Reviewer #2: Yes

Reviewer #3: Yes

Reviewer #4: Yes

6. Review Comments to the Author

Reviewer #2: The authors made all the adjustments that were requested. Thus, the air quality greatly improved. Congratulations to the authors for the excellent work that was done.

Reviewer #3: Reviewer comments

Manuscript Number: PONE-D-20-30645_R1

Title "Predictors of health self-management behaviour in Kazakh patients with metabolic syndrome: A cross-sectional study in China".

Generally speaking:

Thank you for providing me the opportunity to review this manuscript that raises important issues about health self-management behavior in Kazakh patients with metabolic syndrome.

Comment 1:

1. ABSTRACT:

a) A brief background, plus the aim, should be added.

Comment 2:

2. INTRODUCTION:

a) Newest Global/ Regional/ China prevalence of metabolic syndrome should be mentioned.

Comment 3:

3. METHODS:

a) Sample size should be explained.

Comment 4:

4. RESULTS:

a) In the comments of table 3, it is advisable to explain the main influencing factors for the total SMB and for each dimension in the table.

b) Please, revise and redo the regression analysis (table 3) considering the positive and negative influencing factors. For example, for education, the B is -ve and the OR is less than 1, so it is a negative influencing factor not a positive one and so on.

Comment 5:

5. DISCUSSION:

a) It is advisable to explain the study objective at the beginning of the discussion.

b) It is better to write that any factor has OR less than 1 is less likely to cause health self-management behavior and vice versa.

c) The word “Table 3” should not be written in the discussion.

Comment 6:

6. STRENGTHS AND LIMITATIONS:

a) Please, analyze the strengths and limitations of the study.

Reviewer #4: Manuscript Number: PONE-D-20-30645R1

In this manuscript, the authors aimed to identify the factors predicting health self-management behavior among Kazakh patients with metabolic syndrome. Here are my comments:

Abstract:

Please rephrase the following sentence: “The significant positive predictors of health self-management behaviour were gender…”. It is not clear whether male or female was the positive predictor.

Introduction:

Please provide references for the following sentences in the Introduction section:

1. “The occurrence of MS is related to many factors, such as race, genetics, culture, religious beliefs, and lifestyle.”.

2. “Long-term nomadic life has caused Kazakhs to develop a unique lifestyle and cultural background that is distinct from other ethnic groups, which may be the main reason for their high incidence of MS.”

Methods:

Inclusion criteria: should add Kazakhs.

Results:

There appears to be a discrepancy between the text and Table 1 regarding percentage of women, in the table; 52.2% while in the text; 52.8%.

Line 159 “Overweight and obesity were defined as a BMI≥24.0 kg/m2 and a BMI≥28.0 kg/m2, respectively.”, the definition should be in the Methods section.

Figure 1, Table 3 and Table S1– Please provide the expanded form of the abbreviations (in the figure cation and table note).

Lines 208-210: “…gender, education, family monthly income per capita, weight, knowledge of MS, and self-efficacy were positive influencing factors of health self-management behaviour…”. It is not clear if female or male was associated positively with self-management. Please rewrite the sentence.

Discussion:

- Line 234- “The results of the above studies all indicated that the overall level of health…” needs further editing.

- The following paragraph is not evidence-based (at least not by this paper) and it seems to be not correct. The explanation may relate to nomadic unique lifestyle (and not to the religious beliefs). I suggest to delete or rephrase the following:

“Here, the explanation may relate to the religious beliefs of the studied ethnic groups. Religious taboos in the Quran, such as those regarding diet, behaviour and certain types of emotional expression, have a profound influence on the Kazakhs who believe in Islam. These taboos not only play an important guiding role with respect to their health concept, lifestyles and social life, but also have a strong restraining effect, which may affect the health self-management behaviour of Kazakh MS patients to a certain extent.”

- There is no need to specify in detail the results in the Discussion section, it should summarize the main points but not duplicate of the Results section. I suggest to minimize the univariate analysis results in the Discussion section. For example, you can minimize the marked paragraph:

The univariate analysis showed that gender was not significantly associated with the total health self-

management behaviour scores and the scores of each dimension among Kazakh MS patients in China

(Table 1). However, gender was included in the multi-factor regression analysis because we considered that gender may have a certain impact on the health self-management behaviour of Kazakh MS patients from a professional perspective. In the logistic regression analysis, being male was a protective factor for communication with physicians in Kazakh MS patients (Table 3).

-Line 294-295: "Education was a risk factor for exercise management behaviour…" Counterintuitive, education was a risk factor? Please note that the OR in table 3 was not significant.

In the Discussion section please add the limitations of the study, for example, the use of convenience sampling.

7. PLOS authors have the option to publish the peer review history of their article (what does this mean?). If published, this will include your full peer review and any attached files.

Reviewer #2: **Yes: **Mateus Dias Antunes

Reviewer #3: No

Reviewer #4: No

---

## [Author Response · Author response to Decision Letter 1]

27 Sep 2022

Dear Editors and Reviewers:

We would like to thank you for your kind letter and for the reviewers’ constructive comments concerning our manuscript entitled “Predictors of health self-management behaviour in Kazakh patients with metabolic syndrome: A cross-sectional study in China” (Manuscript Number: PONE-D-20-30645R1). All the comments were valuable and highly helpful in revising and improving our paper as well that ensuring that the significance of our research was clear. All the authors seriously discussed all the comments. According to the reviewers’ comments, we have made changes that we hope will meet with your approval. The revised sections are marked in red in the paper. The main corrections in the paper and our responses to the reviewer’s comments are as follows:

Reviewer #2: 

Comment: “The authors made all the adjustments that were requested. Thus, the air quality greatly improved. Congratulations to the authors for the excellent work that was done”.

Response: 

Thank you very much for your positive comments about the further adjustments and improvements made to the manuscript.

Reviewer #3: 

Comment 1: “ABSTRACT: A brief background, plus the aim, should be added.”

Response: 

Considering the reviewer’s suggestion, we have added this information to lines 3-7.

Comment 2: “INTRODUCTION: Newest Global/ Regional/ China prevalence of metabolic syndrome should be mentioned.”

Response: 

Considering the reviewer’s suggestion, we again researched the prevalence of metabolic syndrome and unfortunately did not find the latest reports.

Comment 3: “METHODS: Sample size should be explained.”

Response: 

Considering the reviewer’s suggestion, we have provided this explanation in lines 86-92. 

Comment 4: “RESULTS: a) In the comments of table 3, it is advisable to explain the main influencing factors for the total SMB and for each dimension in the table.”

Response: 

Thank you very much for your suggestion. Indeed, in addition to reporting the main influencing factors of the overall SMB in a comprehensive manner, a further presentation of the influences of each dimension would make the results clearer. However, due to space constraints, we have shown the influencing factors for each dimension in detail in table 3, so no further textual presentation has been made.

Comment 4: “RESULTS: b) Please, revise and redo the regression analysis (table 3) considering the positive and negative influencing factors. For example, for education, the B is -ve and the OR is less than 1, so it is a negative influencing factor not a positive one and so on.”

Response:

Thank you very much for your suggestion. We have checked and analysed this section with care. In logistic regression models, the regression coefficient (B) is logarithmically related to the dominance ratio, so that when the regression coefficient is -ve, it corresponds to an OR<1. The dominance ratio can be used as an indicator to estimate the effect size, which measures the magnitude of the dominant influence of an independent variable, and the significance of the OR value is that when assigning a larger value to the event occurrence group,

(1) OR=1, indicating that OR=1 means that the factor has no effect on the occurrence of the event; 

(2) OR>1 means that the factor is a risk factor for the occurrence of the event (negative influence);

(3) OR<1 means that the factor is a protective factor for the occurrence of the event (positive influence). 

In this study, we used the total health self-management behaviour score and each dimension score as dependent variables, divided the self-management behaviour into two groups using the score index, and performed logistic regression. We assigned the values of self-management behaviour as the dependent variable as follows: good=0, poor=1, as shown in Table S3, and the final results were obtained (Table 3). As an example of emotion management behaviour, the presence or absence of chronic disease comorbidity (no=0, yes=1); showed that B = 0.630, OR = 1.874 >1, indicating that the risk of poor emotion management behaviour among study participants with chronic comorbidities was 1.874 times higher than that for participants without chronic comorbidities, and that chronic comorbidities were negative influencing factors on emotion management behaviour. For this reason, we did not modify this section.

Comment 5: “DISCUSSION: a) It is advisable to explain the study objective at the beginning of the discussion. ”

Response: 

Considering the reviewer’s suggestion, we have provided the explanation in lines 237-245.

Comment 5: “DISCUSSION: b) It is better to write that any factor has OR less than 1 is less likely to cause health self-management behavior and vice versa. ”

Response: 

In conjunction with the explanation of "Comment 4(b)", we did not modify the relevant content.

Comment 5: “DISCUSSION: c) The word “Table 3” should not be written in the discussion.”

Response:

Thank you very much for your suggestion. For the completeness of the content and for the convenience of the reader (the content of the supplementary tables are not shown in the text), we have retained the label "Table" in the discussion section.

Comment 6: “STRENGTHS AND LIMITATIONS: a) Please, analyze the strengths and limitations of the study.”

Response:

Considering the reviewer’s suggestion, we have added the limitations of the study in lines 454-462.

Reviewer #4: 

Comment 1: “Abstract: Please rephrase the following sentence: ‘The significant positive predictors of health self-management behaviour were gender…’. It is not clear whether male or female was the positive predictor.”

Response:

We apologize for neglecting the sex-specific description in the manuscript. We have now provided a modification in line 227.

Comment 2: “Introduction: Please provide references for the following sentences in the Introduction section: 1. ‘The occurrence of MS is related to many factors, such as race, genetics, culture, religious beliefs, and lifestyle.’ 2. “Long-term nomadic life has caused Kazakhs to develop a unique lifestyle and cultural background that is distinct from other ethnic groups, which may be the main reason for their high incidence of MS.”

Response:

Thank you very much for your careful reminder. We have provided the supplementary references in lines 49 and 55.

Comment 3: “Methods: Inclusion criteria: should add Kazakhs.”

Response:

Considering the reviewer’s suggestion, we have added Kazakhs to the inclusion criteria in line 79.

Comment 4: “Results: There appears to be a discrepancy between the text and Table 1 regarding percentage of women, in the table; 52.2% while in the text; 52.8%.”

Response:

We apologize for the incorrect number, and we have corrected it in line 165.

Comment 5: “Results: 159 ‘Overweight and obesity were defined as a BMI≥24.0 kg/m2 and a BMI≥28.0 kg/m2, respectively.’, the definition should be in the Methods section.”

Response:

Considering the reviewer’s suggestion, we have moved the definition of overweight and obesity to the methods section in lines 105-106.

Comment 6: “Results: Figure 1, Table 3 and Table S1– Please provide the expanded form of the abbreviations (in the figure cation and table note).”

Response:

Considering the reviewer’s suggestion, we have supplemented the expanded form of the abbreviations in Figure 1, Table 3 and Table S1 in lines 187-188, 233-234 and Table S1 in that order.

Comment 7: “Results: Lines 208-210: ‘…gender, education, family monthly income per capita, weight, knowledge of MS, and self-efficacy were positive influencing factors of health self-management behaviour…’. It is not clear if female or male was associated positively with self-management. Please rewrite the sentence.”

Response:

Considering the reviewer’s suggestion, we have made this point explicit in line 227 based on the results of the study.

Comment 8: “Discussion: -Line 234- ‘The results of the above studies all indicated that the overall level of health…’ needs further editing.”

Response:

Considering the reviewer’s suggestion, we have further edited and improved this sentence in lines 264-266.

Comment 9: “Discussion: The following paragraph is not evidence-based (at least not by this paper) and it seems to be not correct. The explanation may relate to nomadic unique lifestyle (and not to the religious beliefs). I suggest to delete or rephrase the following:

‘Here, the explanation may relate to the religious beliefs of the studied ethnic groups. Religious taboos in the Quran, such as those regarding diet, behaviour and certain types of emotional expression, have a profound influence on the Kazakhs who believe in Islam. These taboos not only play an important guiding role with respect to their health concept, lifestyles and social life, but also have a strong restraining effect, which may affect the health self-management behaviour of Kazakh MS patients to a certain extent.’”

Response:

Thank you very much for your suggestion. This section has been revised accordingly in lines 283-284, but the discussion on the influencing factor "religious beliefs" has been retained. The reasons for this are follows: on the one hand, the reviewer suggested adding this section during the first round of peer review; on the other hand, the authors have been living in Xinjiang, China, and have experienced first-hand that religious beliefs do have an impact on the health concepts, lifestyles and social life of this population.

Comment 10: “Discussion: There is no need to specify in detail the results in the Discussion section, it should summarize the main points but not duplicate of the Results section. I suggest to minimize the univariate analysis results in the Discussion section. For example, you can minimize the marked paragraph: The univariate analysis showed that gender was not significantly associated with the total health self-management behaviour scores and the scores of each dimension among Kazakh MS patients in China (Table 1). However, gender was included in the multi-factor regression analysis because we considered that gender may have a certain impact on the health self-management behaviour of Kazakh MS patients from a professional perspective. In the logistic regression analysis, being male was a protective factor for communication with physicians in Kazakh MS patients (Table 3).”

Response:

Thank you very much for your suggestion, indeed as you say there is no need to specify in detail the results in the discussion section, which we have summarized in lines 301-302.

Comment 11: “Discussion: -Line 294-295: "Education was a risk factor for exercise management behaviour…" Counterintuitive, education was a risk factor? Please note that the OR in table 3 was not significant.”

Response:

Again, we are deeply apologetic for our carelessness and thank you very much for your prompt and careful suggestion. After double-checking the data in Table 3, it was confirmed that the p=0.092 > 0.05 for the influencing factor education was not statistically significant, so we have made a correction in lines 322-323.

Comment 12: “In the Discussion section please add the limitations of the study, for example, the use of convenience sampling.”

Response:

Considering the reviewer’s suggestion, we have added the limitations of the study in lines 454-462.

We earnestly appreciate the editors/reviewers’ sincere feedback and hope that our changes will meet with their approval. PLOS One is an influential, highly informative journal, and we are grateful to have our article considered for publication. Should additional corrections be needed, please let us know.

---

## [Decision Letter · Decision Letter 2]

14 Nov 2022

Predictors of health self-management behaviour in Kazakh patients with metabolic syndrome: A cross-sectional study in China

PONE-D-20-30645R2

Dear Dr. Jiang,

We’re pleased to inform you that your manuscript has been judged scientifically suitable for publication and will be formally accepted for publication once it meets all outstanding technical requirements.

Kind regards,

Rubeena Zakar, Ph.D

Section Editor

PLOS ONE

Additional Editor Comments (optional):

Reviewers' comments:

Reviewer's Responses to Questions

**Comments to the Author**

1. If the authors have adequately addressed your comments raised in a previous round of review and you feel that this manuscript is now acceptable for publication, you may indicate that here to bypass the “Comments to the Author” section, enter your conflict of interest statement in the “Confidential to Editor” section, and submit your "Accept" recommendation.

Reviewer #2: All comments have been addressed

Reviewer #3: All comments have been addressed

2. Is the manuscript technically sound, and do the data support the conclusions?

Reviewer #2: Yes

Reviewer #3: Yes

3. Has the statistical analysis been performed appropriately and rigorously? 

Reviewer #2: Yes

Reviewer #3: Yes

4. Have the authors made all data underlying the findings in their manuscript fully available?

Reviewer #2: Yes

Reviewer #3: Yes

5. Is the manuscript presented in an intelligible fashion and written in standard English?

Reviewer #2: Yes

Reviewer #3: Yes

6. Review Comments to the Author

Reviewer #2: The authors made all the adjustments that were suggested in the first evaluation and now the article has a better quality and can be published. Thank you very much and congratulations for the excellent work.

Reviewer #3: Reviewer comments

Manuscript Number: PONE-D-20-30645_R2

Title "Predictors of health self-management behaviour in Kazakh patients with metabolic syndrome: A cross-sectional study in China".

Thank you for providing me the opportunity to re-review this manuscript that raises important issues about Predictors of health self-management behavior in Kazakh patients with metabolic syndrome: A cross-sectional study in China.

It seems that all corrections were done.

7. PLOS authors have the option to publish the peer review history of their article (what does this mean?). If published, this will include your full peer review and any attached files.

Reviewer #2: **Yes: **Mateus Antunes

Reviewer #3: No

---

## [Editor Report · Acceptance letter]

16 Nov 2022

PONE-D-20-30645R2 

Predictors of health self-management behaviour in Kazakh patients with metabolic syndrome: A cross-sectional study in China 

Dear Dr. Jiang:

I'm pleased to inform you that your manuscript has been deemed suitable for publication in PLOS ONE. Congratulations! Your manuscript is now with our production department. 

Kind regards, 

on behalf of

Dr. Rubeena Zakar 

Section Editor

PLOS ONE